# Biochar-Mediated Zirconium Ferrite Nanocomposites for Tartrazine Dye Removal from Textile Wastewater

**DOI:** 10.3390/nano12162828

**Published:** 2022-08-17

**Authors:** Shazia Perveen, Raziya Nadeem, Farhat Nosheen, Muhammad Imran Asjad, Jan Awrejcewicz, Tauseef Anwar

**Affiliations:** 1Department of Chemistry, University of Agriculture, Faisalabad 38000, Pakistan; 2Institute of Chemistry, University of Sargodha, Sargodha 40100, Pakistan; 3Department of Mathematics, University of Management and Technology, Lahore 54000, Pakistan; 4Department of Automation, Biomechanics and Mechatronics, Lodz University of Technology, Żeromskiego 116, 90-924 Łódź, Poland; 5Department of Physics, The University of Lahore, Lahore 54000, Pakistan

**Keywords:** adsorption, biochar, BC-ZrFe_2_O_5_ NCs, batch study, response surface methodology

## Abstract

To meet the current challenges concerning the removal of dyes from wastewater, an environmentally friendly and efficient treatment technology is urgently needed. The recalcitrant, noxious, carcinogenic and mutagenic compound dyes are a threat to ecology and its removal from textile wastewater is challenge in the current world. Herein, biochar-mediated zirconium ferrite nanocomposites (BC-ZrFe_2_O_5_ NCs) were fabricated with wheat straw-derived biochar and applied for the adsorptive elimination of Tartrazine dye from textile wastewater. The optical and structural properties of synthesized BC-ZrFe_2_O_5_ NCs were characterized via UV/Vis spectroscopy, Fourier transform Infra-red (FTIR), X-Ray diffraction (XRD), Energy dispersive R-Ray (EDX) and Scanning electron microscopy (SEM). The batch modes experiments were executed to explore sorption capacity of BC-ZrFe_2_O_5_ NCs at varying operative conditions, i.e., pH, temperature, contact time, initial dye concentrations and adsorbent dose. BC-ZrFe_2_O_5_ NCs exhibited the highest sorption efficiency among all adsorbents (wheat straw biomass (WSBM), wheat straw biochar (WSBC) and BC-ZrFe_2_O_5_ NCs), having an adsorption capacity of (mg g^−1^) 53.64 ± 0.23, 79.49 ± 0.21 and 89.22 ± 0.31, respectively, for Tartrazine dye at optimum conditions of environmental variables: pH 2, dose rate 0.05 g, temperature 303 K, time of contact 360 min and concentration 100 mg L^−1^. For the optimization of process variables, response surface methodology (RSM) was employed. In order to study the kinetics and the mechanism of the adsorption process, kinetic and equilibrium mathematical models were used, and results revealed 2nd order kinetics and a multilayer chemisorption mechanism due to complexation of hydroxyl, Fe and Zr with dyes functional groups. The nanocomposites were also recovered in five cycles without significant loss (89 to 63%) in adsorption efficacy. This research work provides insight into the fabrication of nanoadsorbents for the efficient adsorption of Tartrazine dye, which can also be employed for practical engineering applications on an industrial scale as efficient and cost effective materials.

## 1. Introduction

Environmental pollution has been increasing in the current global scenario owing to different toxicants discharged from such industrial sectors as textile, fertilizer, cosmetics and pharmaceutical. The textile industries have been found to be the largest generator of colored effluents because of greater consumption of water during various processing operations. However, 20% of the dye is lost during the process of dying due to poor levels of dye fixation to fiber [1]. It is projected that approximately 7 × 10^5^ tons of dyes are being produced by textile industries annually. The synthetic dyes extensively used in textile industries are non-biodegradable because of their complex aromatic structure. Tartrazine dye, whose International Union of Pure and Applied Chemistry (IUPAC) name is trisodium 1-(4-sulfonatophenyl)-4-(4-sulfonatophenylazo)-5-pyrazolone-3-carboxylate (Figure 1), is a typical synthetic, water-soluble anionic dye. Tartrazine dye has a chromophore (-N=N-) entity in its molecular structure, and is an example of azo dye that is utilized in the textile industry to color wool, cotton, polyamide and silk [2].

The chromophoric azo group in dye molecules is known to pose a serious threat to the biosphere when discharged with waste effluents. The potential health risks associated with dyes are hyperactivity, allergic reactions particularly among asthmatics and those with aspirin intolerance, as well as having a bio-accumulative, carcinogenic and mutagenic nature [3]. Once dyes enter into the environment they are rarely removed; hence, their discharge into water reservoirs results in the disruption of aquatic biota, reduction in sunlight penetration and the retardation of photosynthetic potential of some organisms [4]. Dyes also exhibit toxicity in fish, with a lethal 50% concentration (LC_50_) in test animals of more than 100 mg g^−1^. Therefore, the treatment of wastewater containing dyes is important before its disposal otherwise it is a threat to ecology [5]. Concerns about environmental protection have been stimulated around the globe, prompting researchers and scientists to focus their attention on the remediation of wastewater [3]. As dyes are stable against chemical and biological degradation as well as the conventional treatment, approaches have some limitations, including being cost-prohibitive, time-consuming, producing secondary wastes, membrane fouling, incomplete mineralization and sludge formation [6,7]. Thus, the use of conventional methodologies are not recommended for dye removal, such as filtration, settling and others, as these methods are often inefficient in the removal of this class of pollutants [8]. Therefore, some alternative methods are distinguished by their ability to remove dyes from an aqueous medium, such as membrane separation [9], reverse osmosis [10,11], coagulation [11], oxidative remediation [12] and adsorption [13,14], among others.

Thus, the adsorption of dyes from contaminated water has become a popular topic recently due to its environmental and ecological importance [15]. Adsorption is an effective physico-chemical treatment technique due to ease of design, simplicity of the procedure, environmentally friendly approaches and also has the potential to use activated carbon, biological wastes, mineral oxides or polymer materials for the elimination of contaminants from aqueous solution [7]. The strategy for improving the adsorptive efficacy of industrial effluents is the implication of nanotechnology. In this context, metal nanoparticles are extensively studied owing to their catalytic, chemical, mechanical magnetic and other properties. In addition to a decrease in size, the enhanced surface area improves the capability to react, interact and adsorb other compounds [16]. Biochar is an emerging low-cost carbonaceous material synthesized from cheap agro-waste biomass residue. Its production on an industrial scale is already feasible as its propitious potential for environmental applications attracts considerable interest. Furthermore, biochar is considered a promising candidate for the remediation of inorganic and organic contaminants from water, though the removal efficiency is highly dependent on its chemical and physical attributes [16,17]. The native biochar possesses the limited capability to remove toxicants from water reservoirs, particularly for severely contaminated water. Alternatively, owing to the well-defined porous structure and abundant surface functional groups, biochar displays fascinating properties for the rational design of functional materials. Usually, biochar could be exploited to stabilize and disperse nanoparticles in order to enhance their catalytic reactivity for reactions.

More importantly, biochar-mediated catalysts/adsorbents can exert beneficial impacts to control water pollution when employed for the sequestration of organic pollutants. The biochar nanocomposites are cost-effective and efficient adsorbents to meet the stringent quality criteria of healthy and pure water availability [18,19,20]. Although biochar is an interesting material, it must be modified with metal nanoparticles [21,22] in order to increase its sorption capacity towards dyes and other organic and inorganic pollutants [23,24]. Therefore, many research teams have been working to develop alternative nanoadsorbents that can replace activated carbon in the use for pollution control. In comparison with activated carbons, carbon nanotubes (CNTs), biochar and biochar-based nanocomposites are more attractive because of their favorable thermal/chemical stabilities, high selectivity and structural diversity [25,26]. The reported surface area is (340 m^2^ g^−1^) and porosity (0.21 cm^3^ g^−1^) for biochar pyrolyzed at 600 °C [25,26], while for the activated carbon (AC) of Mango seed (MS) husk, the highest specific surface area is 1943 m^2^ g^−1^ and average pore volume 0.397 cm^3^ g^−1^ [27]. Additionally, the recent progress in their large-scale production makes them better for use as ideal organic and inorganic contaminants nanoadsorbents [28]. Extensive experimental work has been conducted on the adsorption of different dyes onto biochar-mediated nanocomposites, such as Congo Red dye [27], Reactive Red 24 [29], Tartrazine dye [30], Tartrazine dye [31], Reactive Blue 19 dye [32], Cr^3+^ and Cd^2+^ [33], Zn^+2^ and Fe^+3^ [34] and Tartrazine dye [35].

The objective of the present work was to determine the fabrication of biochar ZrFe_2_O_5_ nanocomposites through a facile process using biochar as supportive material for adsorbent materials in the removal of Tartrazine dye from aqueous solutions. The effect of contact time, pH and initial concentration on adsorption characteristics of biochar-based nanocomposites was studied, and the experimental data obtained from the equilibrium studies were fitted to the Langmuir and Freundlich adsorption models. In addition, kinetics of the adsorption process was also studied. The adsorption process optimization was carried out using Response Surface Methodology (RSM). The main goal of the present work is to discover new possibilities of combining biochar from cheaper agro-waste biomass with nanomaterials. Recently emerging ZrFe_2_O_5_ nanocomposites with biochar will be an important step towards wastewater treatment and its purification.

## 2. Experimental

### 2.1. Synthesis of Wheat Straw Biochar

Wheat straw biomass was obtained from Jhang, Pakistan. It was washed using distilled water to eliminate debris and dust particles and allowed to dry in sunlight and in the oven at 60 °C for 24 h. Then, biomass grounded to fine particle size by food processor (Moulinex., Paris, France) was sieved using an Octagon nano-sieve (OCT-DIGITAL 4527-01) to the size of the 200 mm mesh. The sample in the crucible was kept in a muffle furnace at 600 °C for 1 h. The N_2_ gas flow was maintained continuously from the inlet, forcing the pyrolysis fumes to pass via the outlet pipe and was kept underwater in order to avoid the direct discharge of fumes into the air. After 1 h of heating, the furnace was switched off and allowed to cool to room temperature. Then, the biochar was taken out and stored in an airtight bottle for further experiments.

### 2.2. Synthesis of Zirconium Ferrite (ZrFe_2_O_5_) Nanoparticles

The 0.91 g zirconium nitrate (Zr(NO_3_)_4_) and 2.73 g ferric chloride hexahydrate (FeCl_3_.6H_2_O) were mixed into 100 mL of distilled water. The suspension was oscillated ultrasonically for 8 min to avoid zirconium particle conglomeration. The pH of the suspension was maintained at 9.0 by adding 1 N NaOH solution. The suspension was stirred vigorously to avoid the precipitation of zirconium, and the particles were separated by centrifugation after 5 h stirring. The precipitates were washed with distilled water and then placed in a heating oven for 5 h at 160 °C to dry [36].

### 2.3. Synthesis of Biochar Based Nanocomposites

The biochar-supported ZrFe_2_O_5_ NCs was synthesized according to the method described by [20] with little modification. In a 250 mL conical flask, 10 g of sieved biochar and 0.1 g of ZrFe_2_O_5_ nanoparticles were mixed using 150 mL of distilled water under reflux conditions for 24 h. After this time duration, the conical flask was taken out and allowed to settle down for the next 12 h. When the nanocomposites had settled, the supernatant was removed. The obtained composites were dried at 50 °C and grounded into fine powder form.

### 2.4. Characterization of Nanocomposites

The functional group of BC-ZrFe_2_O_5_ NCs was interpreted and analyzed using the FTIR spectrometer (Bruker Tensor 27, Bruker, Hamburg, Germany) (sample prepared as KBr disc), while the surface structure of BC-ZrFe_2_O_5_ NCs was evaluated via SEM (JEOL JMT 300, Chicago, IL, USA). The elemental composition was determined by EDX analysis, and XRD was used to find the crystal nature of composites using Cu K radiations (D8 ADVAHCL, Bruker, Hamburg, Germany).

### 2.5. Point of Zero Charge (pH_pzc_)

The sorbents’ (BM, BC and BC-ZrFe_2_O_5_ NCs) point of zero charge (pH_pzc_) was measured using the salt addition process. To determine pH_pzc_, the series of 50 mL potassium nitrate (KNO_3_) solutions (0.1 M each) were made, and pH was adjusted in a range from 1–12 by adding 0.1 N NaOH and HCl. Then, 0.1 g of adsorbent was added to each solution and was kept for a period of 48 h with intermittent shaking. The final pH (pH_f_) of the solution was noted and the difference between final and initial pH (ΔpH) (*Y*-axis) was plotted versus initial pH (*X*-axis). The intersection point of the curve yield was pH_pzc_ [37].

### 2.6. Preparation of Dye Solutions

Tartrazine dye was utilized in experiments without purification. To prepare 1000 mg L^−1^ stock solution 1 g of dye was dissolved in 1000 mL of deionized water. The solution of various concentrations (ranged from 10–200 mg L^−1^) was prepared by adequate dilution of stock solution for experimental work. Tartrazine dye being anionic had *λ*_max_ 430 nm.

### 2.7. Batch Experimental Studies

Optimization of imperative environmental variables as pH (1–10), sorbent dosage (0.05–0.1 g), contacts time (0–480 min), initial dye concentration (10–200 mg L^−1^) and temperature (303–333 K) for the elimination of Tartrazine dye was conducted via classical approach. The conical flasks (250 mL) comprising 50 mL of the dye solution of known pH, adsorbent dosage and concentration, were shaken in an orbital shaking incubator (PA250/25H) at 200 rpm for 480 min. The pH was adjusted by adding 0.1 N NaOH and HCl solutions. The blank solutions were also carried out under identical experimental conditions, excepting sorbent addition. All experiments were executed in triplicate and findings were determined as mean standard deviation (±SD). After a specified interval of time, the sample solutions were taken out and concentrations of residual dye solutions were determined using UV-Vis spectrophotometer (Schimadzu Corporation, Tokyo, Japan).

Percent removal R% and the equilibrium sorption uptake, q_e_ (mg g^−1^), was calculated using the following relationships: (1)$$\mathrm{Removal}\;\%=\frac{\left(C_{o}-C_{e}\right)\;}{C_{o}}\times 100$$
(2)$$q_{e}=\frac{\left(C_{o}-C_{e}\right)\;V}{W}$$
where C_o_ is the initial dye concentration (mg L^−1^), C_e_ is the equilibrium dye concentration (mg L^−1^), W is the mass of the sorbent (g) and V is solution volume (L).

### 2.8. Optimization of Parameters Using Response Surface Methodology (RSM)

The Response Surface Methodology (RSM) was applied for an interactive study of important influential parameters using “design expert” software. Central Composite Design (CCD) gives an idea about the fitness of experimental data with comparatively lesser numbers of run/design point, thus, reduced overall cost of experiments [38]. The adsorbent dose, pH, contact time and initial dye concentration were selected as independent variables and adsorption capacity (q_e_) was the dependent (response) variable. There was a total of thirty experimental runs, comprising three central points, eight factorial points and six axial points. X_i_ = −1, 0 and 1 were three levels of each input variable [39]. The least-square followed by the second-order differential model was taken into consideration for interpretation of correlation among independent and dependent (response) variables.
Y = β_o_ + ∑^k^_i = 1_ β_i_x_i_ + ∑^k^_i = 1_β_ii_x_i_^2^ + ∑^k^_i = 1_∑^k^_i_ ≠ _i = i_β_ij_x_i_x_j_ + ε(3)
where Y is the adsorption capacity (response), β_o_ belongs to coefficients possessing specific numerical values, β_i_, β_ii_ and β_ij_ are coefficients about linear, quadratic and interaction effects, respectively, while ε (Epsilon) is random error and k is a number of independent variables. The suitability and validity of polynomial equations were monitored to check significance by computing statistically the values of the regression coefficient (R^2^) by means of analysis of variance (ANOVA) and F test at 0.05 probability (*p*) [40].

### 2.9. Kinetic and Adsorption Isotherm Models

Sorption kinetics experiments were carried out by using pseudo-1st order (Lagergren, 1898), pseudo-2nd order (Ho et al., 2000) and intraparticles diffusion (Webers and Morris, 1963) kinetic models.

The Freundlich (Freundlich, 1906) and Langmuir adsorption isotherms were investigated in the current study for exploring the adsorption mechanism in Tartrazine dye removal.

## 3. Results and Discussions

### 3.1. Characterization of BC-ZrFe_2_O_5_ NCs

Point of zero charge (pH_pzc_) is the pH at which charge on a sorbent’s surface is neutral, i.e., the amount of electric positive and negative charges is equal. The pH_pzc_ of wheat straw biomass (WSBM), wheat straw biochar (WSBC) and BC-ZrFe_2_O_5_ NCs was observed to be 3.6, 2.5 and 3.8, respectively, as presented in Figure 2. Below this pH value, the adsorbents attain net positive charge because of functional group protonation resulting in strong electrostatic attractions among anionic dye and adsorbents. Beyond this pH value, the sorbent’s surface acquires a negative charge. Thus, the sorption of anionic pollutants (dye) was preferred at pH < pH_pzc_ where sorbent surfaces become positively charged [23]. These results are in accordance with the findings of [41].

The Fourier Transform Infra-red (FTIR) spectroscopy is an important analytical technique that analyzes vibrations characteristic of each functional group in a molecule. The FTIR spectra of biochar, BC-ZrFe_2_O_5_ NCs and Tartrazine dye-loaded BC-ZrFe_2_O_5_ NCs were explored in a range from 400–4000 cm^−1^ as given in Figure 3b. The FTIR spectra of unloaded BC-ZrFe_2_O_5_ NCs showed the presence of peaks in the range from 3512–3113, 1559, 1384, 1048 and 762 cm^−1^. The broad band from 3512–3113 cm^−1^ was attributed to the O-H vibration of ZrO-H on the surface of the sorbent. The vibration peak at 1595 cm^−1^ indicated the presence of a –C=C– group of aromatic rings. The peak at a region of 1384 cm^−1^ represented the presence of the alkyl group in the structure of BC-ZrFe_2_O_5_ NCs. A distinctive sharp peak at 1061 cm^−1^ was assigned to C–O–C stretching vibrations while the Zr–Fe bond was responsible for vibration at 762 cm^−1^. In the FTIR spectra of biochar (BC), the band from 3371–2825 cm^−1^ was attributed to the O–H group (carboxylic acids, phenols and alcohols) on the surface of sorbents such as cellulose, pectin and lignin. The peak at 1584 cm^−1^ indicated the stretching vibration of the –C=C–group. The peaks at 1184 cm^−1^ embodied the presence of C–O–C functional groups in BC. In the case of spectra of dye-loaded BC-ZrFe_2_O_5_ NCs, there is distinctive vanishing and broadening of some bands owing to their involvement in the adsorption process. Thus, FTIR spectra specified the functional groups and exchanging active sites on which adsorption occurred [42,43].

The X-ray diffraction (XRD) pattern of BC-ZrFe_2_O_5_ NCs before and after dye loading was recorded for the determination of the crystalline nature of adsorbents as shown in Figure 3a. The peaks 2*θ* at position 28.9°, 36° and 40° correspond to (012) (200) (311) and (220) planes for BC-ZrFe_2_O_5_ NCs matched with standard card no. 96-152-3795. These planes could be attributed to a tetragonal crystal structure. The size of the crystal was calculated by using the Debye–Scherrer formula as given below: (4)$$D=\frac{K\lambda }{\beta cos\theta }$$
where in *D* is the size of the crystal, λ is the X-ray wavelength (Cu Kα radiations), *K* is the Scherrer constant, *θ* is the angle of diffraction and *β* is full with half maximum. The size of the BC-ZrFe_2_O_5_ NCs crystal was calculated to be 40.72 nm. There was an obvious change in XRD spectra after dye sorption owing to electrostatic and complex formation being a major mechanism in dye removal. These findings are in agreement to the XRD pattern reported by [18].

The morphological characteristics and surface features, i.e., shape, size, pore properties and arrangements of particles of the adsorbent were studied using a Scanning Electron Microscope (SEM). The higher the number of pores, the higher the dye sorption onto the adsorbent surface. The SEM images of free and Tartrazine dye-loaded wheat straw biochar (WSBC) are shown in SI Appendix A, while the SEM micrographs of free BC-ZrFe_2_O_5_ NCs and Tartrazine dye-loaded adsorbents are depicted in Figure 4a–f at different magnification levels. These images clearly show porous, rough, fibrous, regular and rod-like textures of sorbents facilitating the sorption of dye. In the SEM micrographs, after Tartrazine sorption, the regular rod-like structure seems to be cramped by dye molecules, thus, minimizing the size of pores as well as causing surface roughness, corroborating that the WSBC and BC-ZrFe_2_O_5_ NCs represent suitable morphological profiles for dye take up. These results are related to the reported results of [44] with minor changes due to the difference in experimental conditions.

The Energy Dispersive X-ray (EDX) studies documented the elemental composition of sorbents. The EDX pattern of free and loaded BC-ZrFe_2_O_5_ NCs is depicted in Appendix A. The EDX spectrum verified strong signals for Zr and Fe presence. In addition, Na, Ca, Si, C, O, S, Al and Cl signals were assigned to molecules present in biochar as an integral part. However, in the EDX spectrum of the dye-loaded sorbent, the clear vanishing of Zr, Fe, Na, Ca, C, O and Al and peaks were observed, supporting electrostatic and ionic exchange interaction as a major mechanism for adsorption phenomena. These findings are analogous to the results of [43].

### 3.2. Optimization of Environmental Variables

#### 3.2.1. Impact of pH

Solution pH is a substantial factor that monitors the process of sorption as the pH affects the functional groups’ activity on the adsorbent surfaces. The influence of pH on the remediation of Tartrazine dye from aqueous media is presented in Figure 5a. The sorption was carried out by varying dye solutions pH from 2 to 10 and the findings revealed that the rise in pH culminates in the decline of dye adsorptive removal. The maximum sorptive removal of Tartrazine dye by BM, BC and BC- ZrFe_2_O_5_ NCs was achieved at pH 2. This was attributed to the fact that at acidic conditions, functional groups (binding sites) on adsorbent surfaces get protonated owing to a net increment of a positive charge. The electrostatic attractions among positively charged sorbents and anionic dye surfaces were responsible for the extent of dye elimination from the solution. An increment in medium pH had a contrary influence on sorbent removal capacity for dye because of the increase in negative charges onto the adsorbent surfaces resulting in net electrostatic repulsion among sorbent functional groups and dye anions subsequently causing the reduced sorption of dye [45]. These findings are also in concord to the point of zero charges of BM, BC and BC-ZrFe_2_O_5_ NCs that were found to be 3.6, 2.5 and 3.8, respectively. Thus, below the aforementioned pH values, adsorbents hold net positive charge resulting in greater sorption of anionic Tartrazine dye. Maximum removal (88.09 mg g^−1^) of dye was obtained by utilizing BC-ZrFe_2_O_5_ NCs. These findings were similar to the results reported by [46].

#### 3.2.2. Impact of Contact Time

For cost-effective and large scale wastewater treatment, the contact time course studies between the sorbate and sorbent are imperative in the sorption system. To explain the influence of contact time on the removal of Tartrazine, the experiments were performed by varying time intervals from 15–480 min while keeping the other environmental variables such as pH, concentration, temperature and adsorbent dose constant. Figure 5b illustrates results of time-dependent experimental data points. The outcomes show that during the adsorption process, initially the rate of reaction was rapid, followed by slow removal and, finally, the trend becomes constant on the achievement of equilibrium. Within 90 min, BM, BC and BC-ZrFe_2_O_5_ NCs adsorbents showed up to 60, 71 and 86 mg g^−1^ removal for Tartrazine dye. Approximately 80% of dye uptake took place within 90 min; therefore, this time (90 min) was deemed to be adequate in successive experiments to establish equilibrium. During sorption phenomena, the dye molecules initially come into contact with the boundary of the adsorbents, then adsorb onto the adsorbent surface and finally diffuse into the porous and permeable sorbent that requires longer contact time. Hence, the sorption process arises in two steps, the initial (rapid) stage and final (equilibrated) stage. At the start of the reaction, rapid sorption might be due to the vacancy of all the active binding sites onto the sorbent surface and dye molecules easily occupy the binding sites. As the reaction progressed, the adsorption process slowed down due to the saturation of active sites resulting in the slower movement of adsorbate molecules from the boundary of the adsorbent into the interior of the adsorbent [47,48]. In one study chitosan/geopolymer beads were used for Crystal violet dye sorption and similar results were obtained from the study [24].

#### 3.2.3. Impact of Initial Dye Concentration

The initial dye concentration appears to be of paramount importance as it has a pronounced influence on sorption phenomena. The concentration dependence efficacy of sorbents was recorded by changing Tartrazine concentrations from 10 to 200 mg L^−1^ at pre-optimized conditions of pH (2), sorbent dose (0.05 g) and contact time (120 min). Figure 5c represents the finding regarding the impact of initial dye concentration on sorptive uptake of BM, BC and BC-ZrFe_2_O_5_ NCs. Results revealed that on enhancing the concentration of dye (from 10–200 mg/L) the sorption (mg g^−1^) improved from 5.23–113.24, 5.66–137.88 and 6.69–162.67 for BM, BC and BC-ZrFe_2_O_5_ NCs, respectively. The initial dye concentration offers the main driving force for the collision of all dye molecules between the solid (adsorbent) and aqueous (adsorbate) phases so yielding greater uptake of dye. At lower concentration, the ratio of active sites of the adsorbent surface to available adsorbate molecules was lower; consequently, fractional adsorption no longer dependent upon initial dye concentration. However, at greater concentration, the binding sites accessible for adsorption might be fewer in comparison to the molecules of dye present; therefore, removal strongly depends upon adsorbate concentration. Contrary to this, the % removal decreased on increasing initial concentration because of the saturation of adsorption sites at a fixed sorbent dose [49]. These results are concordant to studies reported by [50] with the analogous trend of increased dye removal with an augmented concentration of dye.

#### 3.2.4. Impact of Sorbent Dosage

The sorbents amount also plays an imperative role in the process of biosorption. To explore the influence of the amount of adsorbent on the removal of Tartrazine, the dose of BM, BC and BC- ZrFe_2_O_5_ NCs was raised from 0.05 to 0.1 g/50 mL of Tartrazine dye solution while keeping initial concentration (100 mg L^−1^), and findings in Figure 5d depicted that the dye uptake was reduced by augmenting the sorbent concentration. Almost 60% reduction in sorption was recorded on enhancing the sorbent and the maximum removal of dye was attained by 0.05 g sorbent dose. It was attributed to the aggregation of particles of sorbents resulting in the lower surface area, hence, the decrease in the active sites to dye molecules ratio [36]. These results closely relate to the literature cited by other researchers [51].

#### 3.2.5. Impact of Temperature

The temperature is an imperious parameter as it is an indicator to determine whether the sorption process is exothermic or endothermic. The experimental data regarding temperature influence on sorption of Tartrazine dye from aqueous solution was studied at a different temperature ranging from 303–333 K as shown in Figure 5e. It was examined from the results that the adsorption of Tartrazine dye onto BM, BC and BC-ZrFe_2_O_5_ NCs was an exothermic process, i.e., rising temperature caused a reduction in dye sorption. This can be elucidated as the rise in temperature weakened binding forces that caused detachment of the dye molecule from the adsorbent’s surface [14]. The highest removal was observed at 303 K with BM, BC and BC-ZrFe_2_O_5_ NCs. It was also examined that the sorption of textile dye onto activated carbon was of exothermic nature [52].

### 3.3. Optimization of BC-ZrFe_2_O_5_ NCs Using Response Surface Methodology

Response Surface Methodology (RSM) was employed for the optimization of process parameters that assisted in identifying the maximum possible interactions among various parameters. Central Composite Design (CCD) of 30 experimental runs was performed to examine the interactive influence of four independent variables including pH, sorbent dose, contact time and the initial concentration of dye for the response (adsorption capacity). To analyze the statistical significance of factors and their interactions, the Fischer’s test for ANOVA was employed. The probability values (*p*-values) for model terms were calculated at a 95% confidence level. *p*-values of each input variable proposed whether it was significant. This multiple regression analysis on experimental data was performed and gave a 2nd order polynomial equation representing the relation among response Y (adsorption capacity) and input variables (pH, dye concentration, contact time and sorbent dose) [38,39]. The summarized results of regression coefficients and ANOVA have been tabulated in Appendix A (for Tartrazine dye).

The model adequacy for the presentation of the results of experimental data was justified by greater values of R^2^ and adj R^2^. The non-significant lack of fit (*p* > 0.05) guarantees the goodness of the suggested model. Various input variables combinations (sorbent dose (g), contact time (min), pH and initial dye concentration (mg L^−1^) were adjusted to perform adsorption experiments using CCD design. Response in terms of adsorption capacity (mg/g) is described with contour and 3D response surface plots (Figure 6 and Figure 7). The response ranged from 11–147 mg g^−1^. The 2nd order regression polynomial equation signified the relation as: Y = 80.9 − 5.67A + 59.29B − 0.93C + 31.83 − 8.21AB + 0.82AC − 2.91AD − 1.50BC + 31.50BD − 1.25CD + 3.39A^2^ − 4.31B^2^ − 5.49C^2^ − 31.49D^2^(5)
where Y = response (adsorption capacity) and A, B, C and D designated input variables, i.e., pH, dye initial concentrations, sorbent dose and contact time, respectively. To assess model significance, the coefficients were calculated. These embody the individual impacts of linear, i.e., A, B, C and D and quadratic terms such as A^2^, B^2^, C^2^ and D^2^ as well as effects of 1st order interactions such as AB, AC, AD, BC, BD and CD. The values of adjusted R^2^ nearer to 1 showed a greater correlation among predicted and observed values [53,54]. It was obvious from the results that all parameters (A, B, C, D) individually as well as in combinations had substantial influence on dye adsorption.

#### Optimization of Input Variables

The two-dimensional (2D) contour plots and corresponding three-dimensional (3D) response surfaces were employed to comprehend the influence of the input variables and optimization on the adsorption process [38]. The findings have been depicted in Figure 6 and Figure 7. The contour and 3D surface plots to examine the mutual impact of concentration and pH have been presented in Figure 6a,b. In these plots, initial concentrations of pH and dye were varied, keeping sorbent amount and contact time constant. As noticed from the plots, the adsorption of Tartrazine dye enhanced with increasing concentration up to a certain limit, but pH had a contrary effect; upon increment in pH, the adsorption of respective dye declined (from 3–10 pH). Very high pH imparts lower adsorption efficiency even if the amount of dye concentration is high due to functional group deprotonation resulting in electrostatic repulsion among sorbent and anionic sorbate [54]. The maximum removal capacity was exhibited at 100 mg/L concentration and 2 pH. In addition, the nearly linear contour plots implied relatively weak interaction among concentration and pH. The interactive influence of adsorbent dosage and pH on dye elimination are displayed in Figure 6c,d. It is evident from the figures that lower levels of sorbent and high levels of concentration support the attainment of a higher removal percentage of dye (while time and concentration were held constant). Higher sorbent concentration causes the aggregation of sorbent molecules, thus, lowering the surface area and active binding sites. The response surface and contour plot due to the combined effect of contact time and pH is shown in Figure 6e,f. It was evident from the graph that time and pH had a paramount impact on the response (sorption of dye). Maximum sorption was detected at lower pH and greater contact time owing to maximum interaction among sorbate and sorbent at greater contact time. The combined effect of contact time and pH on dye removal are described in Figure 7a,b, while keeping initial dye concentration and sorbent dose fixed. The % removal was augmented with increments in contact and reduced with pH that is accredited to the overlapping of binding sites. In addition, from the contour plots, it was evident that elliptical lines implied significant association between contact time and pH. The interaction between contact time and sorbent dose was not so influential (Figure 7c,d); the mutual influence of contact time and sorbent dose has been illustrated in Figure 7e,f. The circular lines of the contour plots signified the interaction among contact time and initial concentration. In order to get higher removal efficacy, we need to maximize contact time and minimize the sorbent dosage. Moreover, the significance of the interaction of the combined effect of pH and sorbent dose could be evident from the linear nature of the contour plots. Combined impacts of input variables in numerical values were also set in terms of the coefficient of the polynomial quadratic equation [54,55].

### 3.4. Adsorption Kinetic Modeling

Kinetic models describe the reaction order of the sorption process based on adsorption capacity of the adsorbent including Lagergren’s first order equation and Ho’s second-order expression. Adsorption kinetic modeling is essential to describe the rate of the transfer of dye from aqueous solution to the surface of the adsorbent, which is important to measure the retention time of the adsorption process necessary for the process of optimization in industry. In the present study, the Tartrazine dye kinetics was investigated at various time intervals to understand the behavior of adsorbents such as WSBM, WSBC and BC-ZrFe_2_O_5_ NCs.

#### 3.4.1. Pseudo-First Order

The rate of reaction is proportional to the empty sites of adsorbents. The pseudo-first order equation is given as: (6)$$\log\left(q_{e}-q_{t}\right)={\mathrm{logq}}_{e}-\frac{K_{1t}}{2.303}$$
where K_1_ (per min) is the adsorption rate constant for the pseudo-first order reaction, q_t_ (mg g^−1^) is the amount of adsorption at time “t”, q_e_ (mg g^−1^) is the amount of adsorption of dye at equilibrium. The K_1_ and q_e_ values were attained from the plot of log (q_e_ − q_t_) verses t. The slopes and intercepts of plots of log (q_e_ − q_t_) verses t was used to compute the K_1_ and q_e_. In Figure 8a, the nonlinear plot of Log (q_e_ − q_t_) versus t represented a non-significant regression coefficient (R^2^) which indicated that pseudo-first order kinetics might be inadequate to understand the adsorption mechanism for Tartrazine dye. Moreover, [56] investigated the biosorption kinetics and equilibrium of RO 22 using peels and fruits of *Trapa bispinosa* and observed that R^2^ values for the pseudo-first order kinetic plots were lesser than pseudo-second order kinetic plots.

#### 3.4.2. Pseudo-Second Order Equation

In the pseudo-second order kinetic model, the rate of adsorption is proportional to the square of number of the empty sites of the adsorbent, i.e., it is based on the adsorption capacity of the adsorbent. The pseudo-second order differential equation is given as: (7)$$\frac{{\mathrm{dq}}_{t}}{t}=K_{2}\;(q_{e}-q_{t}{)}^{2}$$

By applying and integrating boundary conditions q = 0 to q = q_t_ and t = 0 to t = t the pseudo-second order kinetics model may be expressed as: (8)$$\frac{t}{q_{t}}=\frac{1}{K_{2}q^{2}{\;}_{e}}+\frac{1}{q_{e}}$$
where K_2_ (mg g^−1^ min^−1^) is the adsorption rate constant for the pseudo-second order reaction. The value of K_2_ and q_e_ can be calculated from the intercept and slope of the plot of t/q_t_. In Figure 8b, the plots of t/q_t_ versus t showed a linear relationship with a higher regression coefficient (R^2^) than the pseudo first-order model. Kinetics parameters are shown in Appendix A showing chemical adsorption as the rate limiting step for the mechanism of adsorption. Data presented shows there is an acquiescence between pseudo-second order theoretical and experimental q_e_ values for Tartrazine dye. Moreover, the values of R^2^ for the pseudo-second order was almost equal to 1 and significantly higher than the pseudo-first order, suggesting that the system of adsorbents is well designated by the pseudo-second order kinetic model. All the evidence predicted that the adsorption of Tartrazine dye follows the pseudo-second order model that is based on the statement that adsorption may be a rate limiting step. The isotherm depicted that how the molecules were distributed between the liquid and solid phases at equilibrium [57,58].

#### 3.4.3. Interparticles Diffusion Model

The solute molecules transfer from aqueous media to sorbent surfaces in several steps, as continuously fast stirring is involved in the batch study, thus, the rate monitoring step might contain intra-particles or film diffusions or both mechanisms. The intraparticles diffusion model equation is given as: q_t_ = K_pi_t^1/2^ +C_i_(9)
where C_i_ = the intercept describing boundary layer thickness and K_pi_ (mg/g min)^1/2^ represents the rate constant for intraparticle diffusion. The plot q_t_ versus t^1/2^ could be linear only if intraparticle diffusion is involved in the sorption phenomenon, but if the line passes through the origin, then the intraparticle diffusion could be the rate determining step. When the plot did not pass via origin, it indicated boundary layer control at some extent and it showed that the intraparticle diffusion was not the only rate controlling step, but other kinetics models might also control the rate of sorption, all of which could operate simultaneously [32,57,59]. The findings of the intraparticle diffusion model are tabulated in Appendix A. The lower values of R^2^ as compared to the 2nd order kinetic model indicate that Tartrazine adsorption did not follow this model.

### 3.5. Adsorption Equilibrium Studies

Adsorption equilibrium models are used to examine the adsorption mechanism or the interaction between adsorbate and adsorbent at equilibrium and used to deduct the maximum adsorption capacity for adsorbents.

#### 3.5.1. Langmuir Isotherm

The Langmuir isotherm model is effective for the monolayer adsorption process with a limited number of energetically equivalent identical active sites. According to this model, the maximum adsorption takes place when the monolayer of adsorbent was saturated with adsorbate molecules and there is no passage for molecules of adsorbate to pass over the surface of the adsorbent. The maximum monolayer adsorption capacity (q_e_ mg/g) and other factors were examined by the following equation: (10)$$\frac{C_{e}}{q_{e}}=\frac{1}{X_{m}K_{L}}+\frac{C_{e}}{X_{m}}$$
where q_e_ (mg g^−1^) is equilibrium sorption capacity, C_e_ (mg L^−1^) is concentration of dye at equilibrium, X_m_ (mg g^−1^) is complete monolayer or adsorption capacity, K_L_ is Langmuir constant as apparent energy of adsorption. A non-linear plot was found when C_e_/q_e_ is plotted against C_e_. Thus, a non-linear relation between C_e_/q_e_ and C_e_, and the lower value of R^2^ predicted that the adsorption of Tartrazine dye and does not follow the Langmuir isotherm model that is based on the statement that biosorption may be a monolayer. The Langmuir isotherm could also be represented as a dimensionless constant partitioning element for the equilibrium parameters, R_L_ calculated by equation: (11)$$R_{L}=\frac{1}{1+{\mathrm{bC}}_{o}}$$
where C_o_ = initial adsorbate concentration and b = Langmuir constant. R_L_ values indicated isotherm type; favorable (0 < R_L_ < 1), irreversible (R_L_ = 0), unfavorable (R_L_ > 1) or linear (R_L_ = 1) as shown in Figure 9a [56].

#### 3.5.2. Freundlich Isotherm

The Freundlich isotherm is based on the concept that adsorption occurs on heterogeneous surfaces and the adsorption capacity depends on the concentration of dye at equilibrium. The Freundlich equation is an empirical equation, and its linearized form can be given as: (12)$$\mathrm{Log}\;q_{e}=\log\;K_{F}+\frac{1}{n}\log\;C_{e}$$
where the K_F_ (mg g^−1^) and 1/n indicates adsorption capacity and adsorption intensity, respectively. The Freundlich constants KF and 1/n can be calculated from the slope and intercept of the linear plot, with log q_e_ versus log C_e_ as shown in Appendix A. The value of 1/n less than one indicated the simple separation of dye from aqueous solution. The lesser value of 1/n showed that the adsorption was better and a comparatively stronger bond was developed between the adsorbent and adsorbate. The good depiction of adsorption equilibrium by the Freundlich model recommended that adsorption of multilayers occurred for Tartrazine dye which predicts many active sites. The results showed that Freundlich isotherm was best fitted to Tartrazine dye than the Langmuir model [56,59,60].

### 3.6. Adsorption Mechanism

Based on the results of FTIR, EDX, SEM and XRD and the adsorption models in this study, the adsorption mechanism can be proposed. The surface hydroxyl, carbonyl and Zr-Fe groups on BC-ZrFe_2_O_5_ NCs are the active adsorption sites for dye in solution, which may interact by complex formation or ion-exchange with dye molecules. The mechanism of Tartrazine dye adsorption by the BC-ZrFe_2_O_5_ NCs be elucidated as follows: MOH + H^+^↔MOH_2_^+^ ≡ MOH_2_ + TD^−^↔ ≡ MTD + H_2_O ≡ MOH + TD^−^↔ ≡ MTD + OH^−^
where ≡M represents the surface of the adsorbent and TD represent Tartrazine dye.

The ZrFe_2_O_5_ NCs, which are entrapped with the biochar matrix, can form an aqua complex with water and develop a charged surface through amphoteric dissociation. When the pH of the medium is acidic, positively charged surface sites are developed, which attract the negatively charged Tartrazine dye (anionic dye) by electrostatic attraction, resulting in the enhanced dye removal in acidic pH values. With a neutral pH of the solution, dye adsorption can be due to the ion-exchange and complexation reaction between anionic dye and hydroxyl, carbonyl and Zr-Fe ions. The adsorption mechanism has been reported as being able to remove dyes with high selectivity and adsorption capacity The excess hydroxyl ions will compete with dye anions on the active adsorption sites, which lead to the decrease in the adsorption capacity with increasing solution pH.

Based on the FTIR analysis, the complex formation mechanism would play an important role in the dye removal. The hydroxyl groups on the surface of the BC-ZrFe_2_O_5_ NCs could form metal complex dye ions during the removal process. Specifically, at high solution pH, the surface of the adsorbent becomes negatively charged, and stronger electrostatic repulsion between active sites and dye exists. The dye removal would be mainly due to complexation by coordination of Fe and Zr with the dyes’ functional groups.

### 3.7. Catalyst Regeneration and Comparison

Catalyst stability, recyclability and reusability are very important for commercial-scale application. After completion of adsorption, the synthesized BC-ZrFe_2_O_5_ NCs catalyst was separated by centrifugation and washed by using 0.1 N solution of NaOH, then deionized with distilled water and desiccated in the oven at 80 °C and saved for subsequent reaction. It was also seen that the catalytic activity of biochar-based composite remained constant up to five cycles of operation so the recycled BC-ZrFe_2_O_5_ NCs could be reused at least five times with no significant loss of catalytic activity as given in Figure 10a. The removal efficacy declined from 89–63%. This sorbent’s membrane fouling owing to dye sorption could be recovered by centrifuging the sorbents in 1 N solution of NaOH at the completion of each cycle. The larger portion of sorbent could be recovered that was useful as well as economic for the regeneration of sorbents for practical applications [24]. The removal efficacy of sorbents viz. WSBM, WSBC and BC-ZrFe_2_O_5_ NCs in batch experimental mode have been depicted in Figure 10b. The findings revealed that sorption efficacy (mg g^−1^) of WSBM, WSBC and BC-ZrFe_2_O_5_ NCs was 67, 75 and 89 for Tartrazine dye, respectively. The extraordinarily high remediation potential of BC-ZrFe_2_O_5_ NCs for dye was attributed to its larger surface area, smaller size, greater surface functional groups and lower resistance of mass transfer [57,61]. The comparison of current study with already reported research work has been compiled in Table 1.

## 4. Conclusions

The current research aimed to appraise the sorption efficacy of biochar-mediated ZrFe_2_O_5_ nanocomposites for the remediation of Tartrazine dye containing textile wastewater. Batch mode experimentations were executed for this purpose. BC-ZrFe_2_O_5_ NCs depicted high sorption potential for Tartrazine dye. The Response Surface Methodology was used for the optimization of process parameters. For future practical applications of biochar-based nanocomposites on an industrial scale as efficient and cost effective materials, much work is required. Regarding these aspects, very little literature is available, thus, further studies are required. The application of biochar-mediated nanocomposites is usually concentrated at the laboratory scale only, however, its application for treatment of simulated and real wastewater is still lacking. Real wastewater commonly consists of diverse contaminants, hence, resource recycling and selective elimination of the pollutants is of great importance. The main goal of the present work is to discover novel, efficient and ecofriendly nanomaterials for application in wastewater treatment and new possibilities of combining biochar from cheaper agro-waste biomass with nanomaterials.

## Figures and Tables

**Figure 1 nanomaterials-12-02828-f001:**
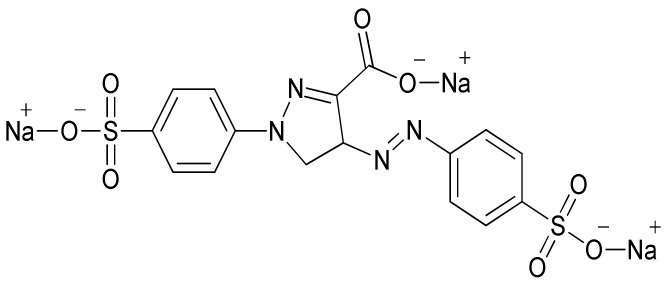
Tartrazine dye structure.

**Figure 2 nanomaterials-12-02828-f002:**
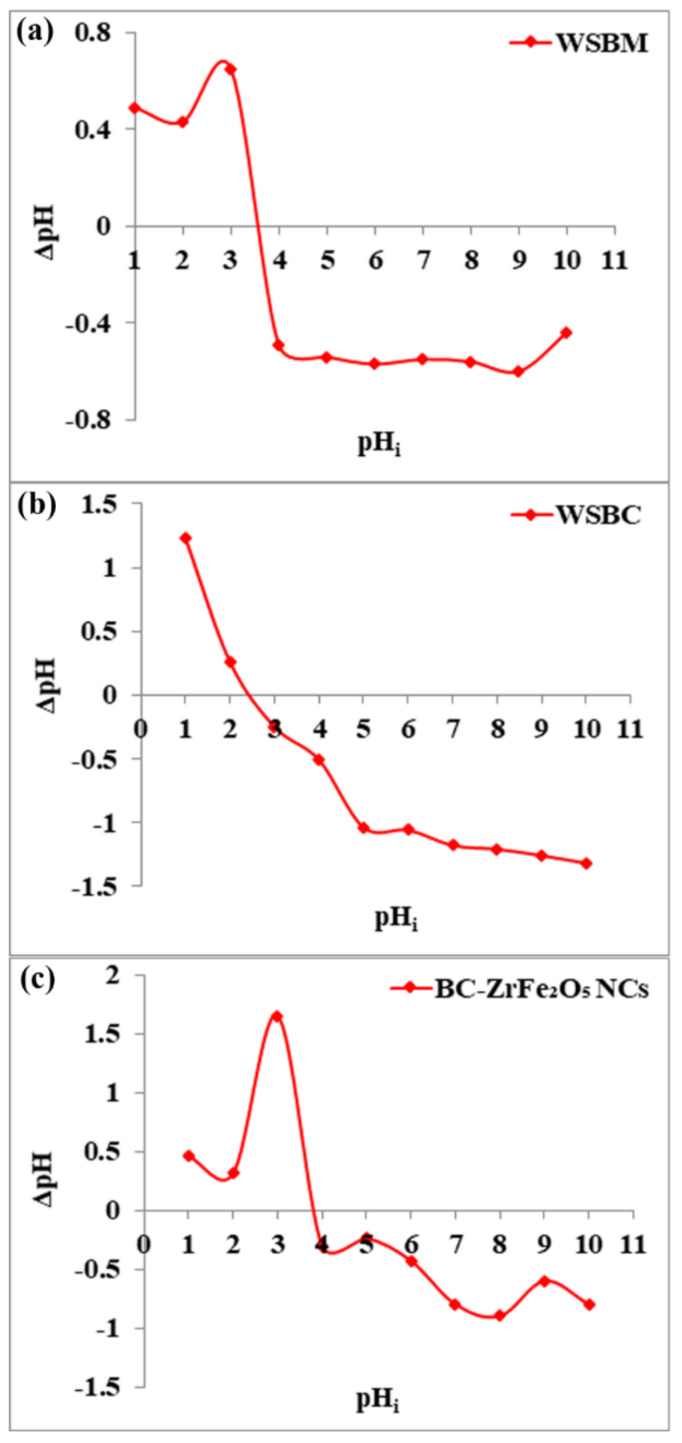
pH_pzc_ of adsorbents: (**a**) wheat straw biomass (WSBM), (**b**) wheat straw biochar (WSBC) and (**c**) biochar-based zirconium ferrite nanocomposite (BC-ZrFe_2_O_5_ NCs).

**Figure 3 nanomaterials-12-02828-f003:**
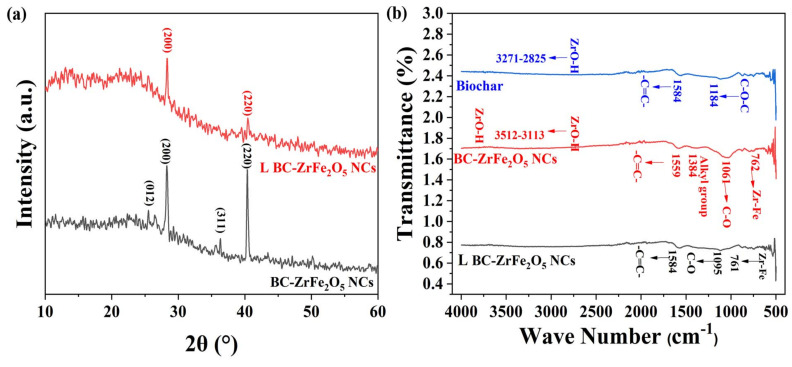
(**a**) XRD pattern of native BC-ZrFe_2_O_5_ NCs and dye-loaded BC-ZrFe_2_O_5_ NCs, (**b**) FT-IR spectrum of Biochar, BC-ZrFe_2_O_5_ NCs and dye-loaded BC-ZrFe_2_O_5_ NCs.

**Figure 4 nanomaterials-12-02828-f004:**
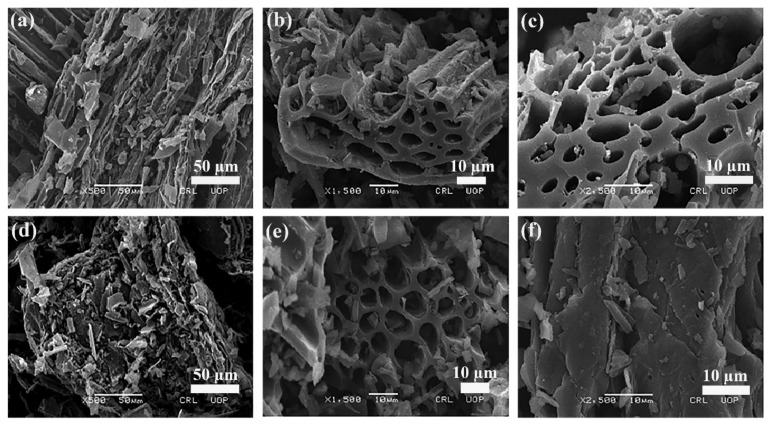
SEM micrographs of BC ZrFe_2_O_5_ NCs (**a**–**c**) and Tartrazine dye-loaded BC-ZrFe_2_O_5_ NCs (**d**–**f**) at three different magnification levels.

**Figure 5 nanomaterials-12-02828-f005:**
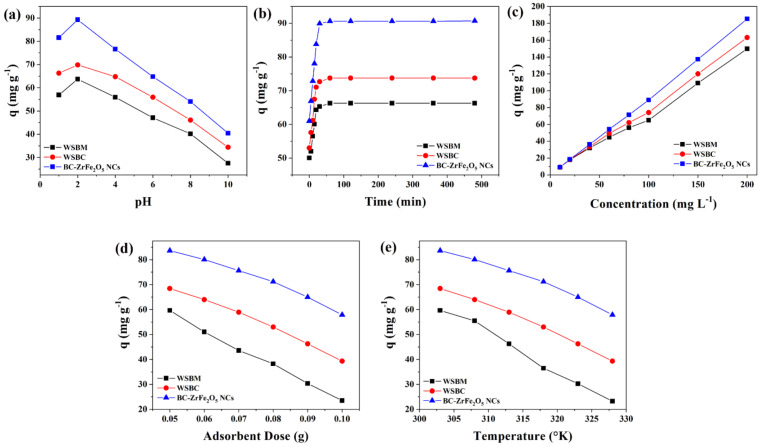
Influence of (**a**) pH, (**b**) time of contact, (**c**) initial concentration, (**d**) sorbent dosage, (**e**) temperature on the adsorption of Tartrazine dye using, WSBM, WSBC and BC-ZrFe_2_O_5_ NCs.

**Figure 6 nanomaterials-12-02828-f006:**
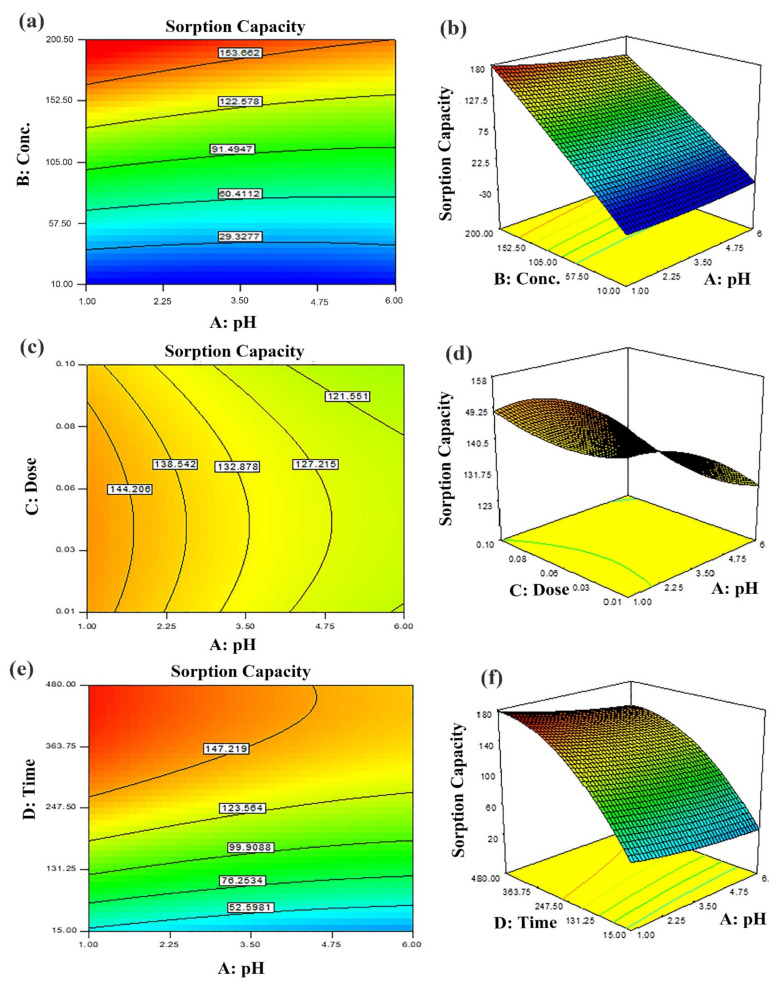
Predicted dye removal contour plots under influence of: (**a**) pH–initial concentration, (**b**) pH–sorbent dose, (**c**) contact time–pH and the related response surface plots in (**d**–**f**) at central point values of other parameters.

**Figure 7 nanomaterials-12-02828-f007:**
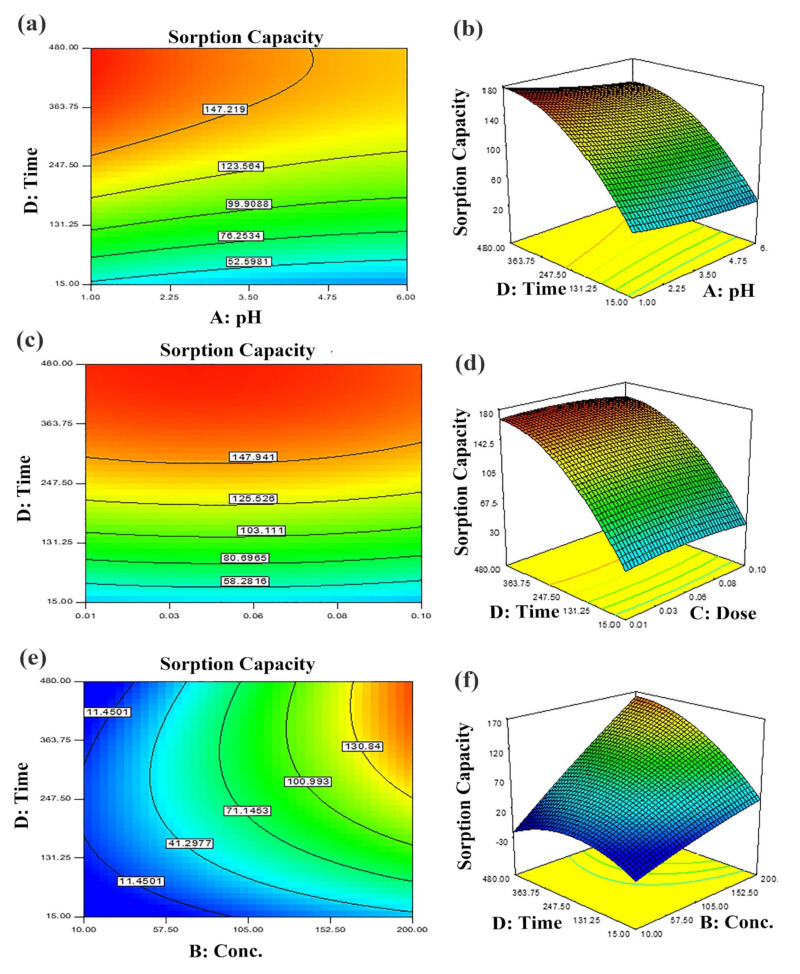
Predicted dye removal contour plots under influence of: (**a**) pH–contact time, (**b**) contact time–sorbent dose and (**c**) contact time–initial concentration and the related response surface plots in (**d**–**f**) at central point values of other parameters.

**Figure 8 nanomaterials-12-02828-f008:**
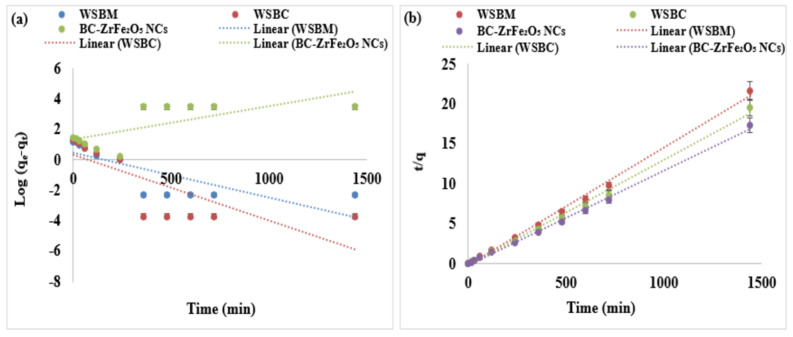
Adsorption kinetic models: (**a**) first order, (**b**) second-order plots of adsorption of Tartrazine dye by WSBM, WSBC, and ZrFe_2_O_5_ NCs.

**Figure 9 nanomaterials-12-02828-f009:**
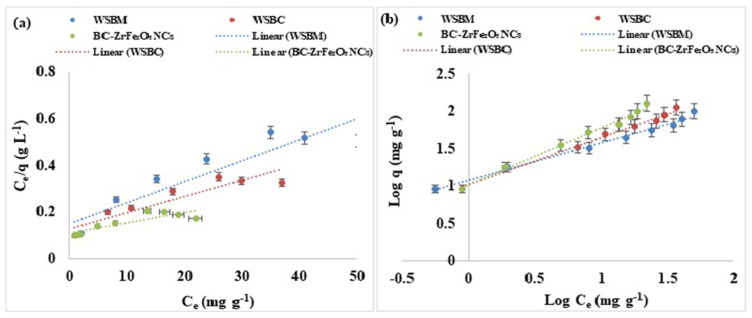
Adsorption equilibrium models: (**a**) Langmuir, (**b**) linearized Freundlich of adsorption of Tartrazine dye by WSBM, WSBC, and BC-ZrFe_2_O_5_ NCs.

**Figure 10 nanomaterials-12-02828-f010:**
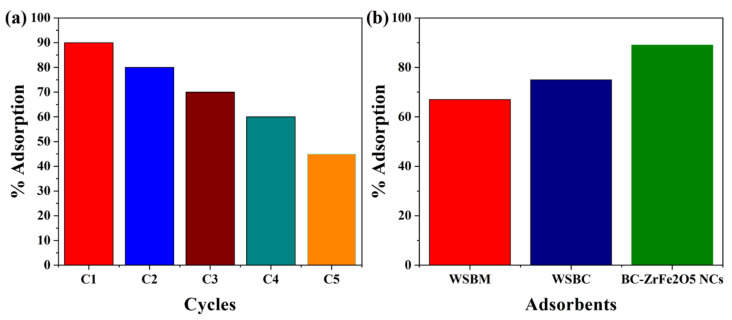
(**a**) Catalyst recyclability and (**b**) comparison among different catalysts (WSBM, WSBC and BC-ZrFe_2_O_5_ NCs) for sorption of Tartrazine dye (at conditions 100 mg g^−1^ dye concentration, 30 °C temperature, 0.5 g sorbent dose).

**Table 1 nanomaterials-12-02828-t001:** Comparison of BC-ZrFe_2_O_5_ NCs to previously reported nano-sorbents for different contaminants.

Adsorbent	Pollutant	Removal Capacity	References
Agro-waste-derived biochars impregnated with ZnO	As(III)	25.9 mg/g	[25]
	Pd(II)	25.8 mg/g	
ZnO/biochar composites	Sulfamethoxazole	90.8%	[26]
	Methyl Orange	88.3%	
ZnO/cotton stalks biochar	Congo Red dye	89.65%	[27]
Agricultural waste-derived biochar modified with ZnO nanoparticles	Reactive red 24	71%	[29]
Polystyrene/magnetite nanocomposite (PS-DVB/Fe_3_O_4_)	Tartrazine dye	90%	[30]
Iron-loaded natural zeolite (NZ-A-Fe)	Tartrazine dye	90%	[31]
Biochar caged zirconium ferrite nanocomposites	Reactive Blue 19 dye	88.8%	[32]
Layered double hydroxide-based material	Cr^3+^	56.95 mg/g	[33]
	Cd^2+^	198.34 mg/g	
Etidronic acid-functionalized layered double hydroxide	Zn^2+^	281.36 mg/g	[34]
	Fe^+3^	206.03 mg/g	
Biochar-supported zinc oxide (BC-ZnO NCs) and Graphene oxide/zinc oxide (BC-GO/ZnO NCs)	Tartrazine dye	78.85 mg/g (87%) 88.8 mg/g (94%)	[35]
ZnO modified with nanobiochar	Phenol	99.8%	[59]
α-MnO_2_ photocatalyst	RhB Organic Dye	95–100%	[62]
Biochar-mediated zirconium ferrite Nanocomposites	Tartrazine dye	89.22 mg/g	**This study**

## Data Availability

Not applicable.

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
