# Peer review of "Biochar-Mediated Zirconium Ferrite Nanocomposites for Tartrazine Dye Removal from Textile Wastewater"

_nanomaterials, 2022, doi:10.3390/nano12162828_

Round 1

Reviewer 1 Report

The Authors managed to improve the quality of the Manuscript according to the Reviewer's comments.

Author Response

Thank you very much for your positive review.

Reviewer 2 Report

In this work BC-ZrFe2O5 NCs were prepared for dye removal from textile wastewater. The revised manuscript is publishable after minor revision:

1)     Figures are in low quality and should be improved

2)     Table 1: Comparison of BC-ZnO and BC-GO/ZnO NCs to previously reported nanosorbents for different contaminants, why BC-ZnO and not BC-ZrFe2O5? Is it an error ?. Authors could compare the results with other nanosorbent  based on Fe2O3, etc…

3)      

Reviewer 3 Report

I have read the manuscript entitled “Biochar mediated Zirconium-Ferrite Nanocomposites for Tartrazine Dye removal from textile wastewater” by T Anwar et al submitted to Nanomaterials MDPI. Industrial wastewater disposals from textile, paper, pharmaceutical, dye, and dye intermediate industries mostly contain both organic dyes as well as inorganic heavy metal ions which are highly toxic and non-biodegradable. Natural dyes are environmentally and economically far better to that of currently available ruthenium-based dyes due to its toxicity and cost factor. The importance of water in the sustenance of life cannot be overstated as it is present in high amounts in all biomass. Heterogeneous catalysis also helps to minimize wastes derived from reaction workup, contributing to the development of green chemical processes. Therefore, the reported work is significant area of research. The paper describes biochar mediated Zr-Ferrite nanocomposites with good characterization and dye reduction activity. The reported work appears to be relatively new. The work is fairly well presented, and manuscript is well written with reasonable characterizations. However, some revision is required before rendering a final decision.

My specific points are below:

·         In Page 1, is it 7 times 10 to the power of 5 tons?

·         On pages 2 -3, the last paragraph on page 2, and the first two paragraphs on Page 2: Biochar is common as an adsorbent due to its surface area, porous nature, and large amounts of surface functional groups. Please mention the surface area and how it differs from Activated carbon reported in the literature derived from agricultural waste (ChemPlusChem 87 (2022) e202200126; doi.org/10.1016/j.ceja.2021.100158)

·         Where the wheat straw biomass has been obtained?

·         What is the mechanism of dye adsorption proposed here with a functional group perspective?

·         How the current study compares to that of RhB organic dye reported in the literature? Please refer to this article (ChemistrySelect 1 (2016) 4277 – 4285) and cite it in terms of the absorption coefficient.

·         Does the surface area ensure the greater availability of active surface/sites for tartrazine uptake?

·         How does its function in electrical neutrality in aqueous media (neutral pH)?

Round 2

Reviewer 3 Report

In this reviewer’s opinion, the revised version is OK. In which, the authors have taken my queries into a consideration and revised the manuscript.  

This manuscript is a resubmission of an earlier submission. The following is a list of the peer review reports and author responses from that submission.

Round 1

Reviewer 1 Report

In this work, novel biochar mediated zirconium ferrite nanocomposites were fabricated with the assistance of wheat straw derived biochar and applied for the adsorptive elimination of Tartrazine dye from textile wastewater. This work is praiseworthy for its rigorous experimental data and wealthy theoretical explanations. Recommendation: Major Revision

1.      Abstract: The main content in the abstract should include important conclusions and shining results of this study, including the representative data, the underlying mechanisms and the future enlightenment, instead of describing the detailed experimental work of this study.

2.      Experimental: Much information can be viewed on the Internet or introduced in similar articles. The author should simplify the introduction or put it in supplementary materials.

3.      Authors should provide the economic study of the production for commercial viability. It is suggested that the author briefly explain what enlightenment this work has for other researchers.

4.      All adsorption experiments should be repeated three times, and error bars should be added to the figure.

5.      It is suggested that the author should add some articles about new adsorption materials in the introduction. The flowing clay mineral is a good adsorbent for contaminants and should compare with this work. (10.1016/j.jhazmat.2021.128062, 10.1021/acsami.1c22035).

6.   What was the pH of the solution at equilibrium?

7. The language needs to be further improved.

Reviewer 2 Report

Anwar and co-workers have presented work on dye adsorption/removal by a Biochar Mediated Zirconium-Ferrite Nanocomposites. The work is nice and could be accepted for publication after addressing the comments given below.

1.     In the EDS spectrum of dye-loaded BC-ZrFe2O5 NCs, the peak for Zr and Fe seems to be missing. Does it indicate that BC-ZrFe2O5 NCs are not water stable and tetrazine is getting adsorbed only by the BC?

2.     In section 3.2, subsection 3.2.1 Authors said that ‘The influence of pH on the remediation of Tartrazine dye from aqueous media is presented in Fig. 5a’ However Fig. 5a could not be located in the MS. It is suggested to verify the assigned Figure numbers again.

3.     The adsorption of any dye can be visualized through the necked eyes and through UV-vis absorption spectroscopy. Authors are suggested to include the pictures of the dye adsorption process and their corresponding UV-vis spectrum. Furthermore, there are many dyes which are released from the textile industry, can the authors shed a light on why only tetrazine dye was selected? Tetrazine is a negative dye, did the authors try to use any positive dye and tried to see the adsorption behaviour of BC-ZrFe2O5 NCs?

4.     To attract the broader reader, authors are also advised to add a comparison table and elaborate on the emphasis as to why the biochar Mediated Zirconium-Ferrite Nanocomposites is better adsorbent for tetrazine than the reported ones.

5.     For the catalysts cycle, authors are suggested to give pictures of each cycle (before and after the dye adsorption process) 

Reviewer 3 Report

The manuscript aimed on the preparation, characterization and application of biochar mediated zirconium ferrite nanocomposites (BC-ZrFe2O5 NCs) for the adsorptive elimination of Tartrazine dye from textile wastewater. The manuscript is well organized and the results are interesting for the reader of this journal. I recommend major revision before possible publication of this work:

1)     In the abstract, authors could present the novelty of this work to add values for the obtained results and materials.

2)     The introduction is poor and need more discussion on the materials used for water treatment.

3)     Authors could discuss more previously work in the introduction (only 9 reference are not sufficient to provide the necessity of this work).

4)     In the results and discussion, authors could summarize and reduce the high number of subtitle (2.1: 2.1.1; etc…) in the manuscript.

5)     A comparative study should be added to confirm the novelty of this work on water treatments.
